# An Infinity Tube with an Expansion Chamber for Noise Control in the Ductwork System

**DOI:** 10.3390/s23010305

**Published:** 2022-12-28

**Authors:** Rong Xue, Cheuk Ming Mak, Chenzhi Cai, Kuen Wai Ma

**Affiliations:** 1Department of Building Environment and Energy Engineering, The Hong Kong Polytechnic University, Hung Hom, Kowloon, Hong Kong; 2School of Civil Engineering, Central South University, Changsha 410017, China

**Keywords:** infinity tube, transfer matrix method, transmission loss, noise control

## Abstract

This paper proposes a muffler with simple geometry to effectively reduce low-frequency noise in ductwork systems. A muffler named infinity tube with an expansion chamber (ITEC) is developed from the infinity tube (IT). Theoretical and numerical analyses of wave propagation in the ITEC have been conducted in this paper. The transfer matrix method is adopted to predict transmission loss theoretically. The theoretical results are validated by the finite element method simulation. The comparison of the transmission loss between the IT and ITEC illustrates that the ITEC has an advantage in low-frequency noise reduction. The transmission loss results of the ITEC are compared with the Helmholtz resonator system to assess the potential for industrial application. Finally, the geometric parameters of the proposed ITEC on its noise attenuation performance have been analyzed. The proposed ITEC can effectively reduce low-frequency noise, and it is suitable for ductwork systems in constrained spaces.

## 1. Introduction

The ductwork system is of vital importance to modern buildings [1]. It is an essential part of the HVAC system to supply fresh air, recycle exhaust, and maintain a comfortable indoor environment. However, the ductwork system always encounters noise problems [2,3]. Researchers have invented many kinds of well-designed mufflers to reduce duct-borne noise in ductwork systems. In industry, dissipative and reactive silencers have a wide range of applications [4]. The principle of dissipative silencers is using sound-absorbing materials to convert sound energy into heat. However, subject to the dimensions of porous material, the dissipative silencers are not suitable for attenuating low-frequency noise [5]. The reactive silencers are composed of acoustic elements which could alter the impedance and reflect the incident acoustic waves. The Helmholtz resonator [6], quarter-wavelength tube [7], and expansion chamber [8] are typical reactive silencers used in duct systems. For the Helmholtz resonator and quarter-wavelength tube, the noise attenuation band is near the resonance frequency and relatively narrow. The expansion chamber needs a large expansion ratio to maintain the noise attenuation ability, which leads the expansion chamber to be cumbersome [9]. A novel muffler is required to avoid the disadvantages of the existing silencers.

The Herschel-Quincke tube (hereafter HQ) consists of two pipes parallelly mounted along with arbitrary length and cross-section area [10,11]. Stewart discussed the sound transmission characteristic of HQ tube devices theoretically [12]. After years of research, the HQ tube system has been shown to be an effective silencer for low-frequency noise attenuation. Selamet et al. [13] conducted an experimental, theoretical, and computational investigation on HQ tube transmission loss (TL). Thereafter, Selamet et al. [14] eliminated the geometric restrictions of HQ and altered the HQ tube to an N-duct configuration. Kim et al. [15] designed a virtual HQ tube system to achieve the desired transmission loss performance under a required frequency range. Wang et al. [16] combined the HQ tube with micro-perforated panels and developed a new noise control device. Ahmadian et al. [17] developed a genetic algorithm to optimize the parameters of a HQ tube. Mazzaro et al. [18] numerically investigated air flow movement inside the HQ tubes. The disadvantage of the HQ tube is clear. HQ needs two length parameters to locate the position on the main duct and determine the resonant frequency [19]. Therefore, compared with other reactive silencers, HQ appears to be cumbersome. In addition, the research by Torregrosa et al. [20] showed that the HQ tube would cause flow repartition between the main duct and HQ device.

Lato et al. [19] developed the traditional HQ tube into the infinity tube (hereafter IT) by combining the previous two connecting points of the HQ in different duct cross-sections into one cross-section. With a simpler geometry, the IT is easier to manufacture and install than the HQ tube. In addition, IT could avoid flow repartition. The research showed that IT is an innovative muffler and would have the potential for industrial applications. To further improve the noise attenuation ability of IT, an expansion chamber muffler would be used to replace the side branch of the IT device. The improved IT device would be referred to as the infinite tube with an expansion chamber (ITEC) throughout the whole study.

In this paper, the transfer matrix method (TMM) is adopted to predict the noise attenuation ability of the ITEC. TMM and the statement of pressure equality and conservation of volumetric flow are performed to solve the transmission loss of the ITEC analytically. The finite element method (FEM) simulation of the ITEC has been conducted to validate the theoretical prediction results. Then, the transmission loss results of the ITEC are compared with infinity tubes and other silencers to examine the noise attenuation performance. The most frequently used reactive silencer in industry, the Helmholtz resonator, is chosen for comparison. Finally, the effects of the geometric parameter of the proposed ITEC on the noise attenuation performance are investigated.

## 2. Analytical Model of the ITEC

### 2.1. Sound Propagation Inside the Duct System and Transfer Matrix Method

Considering only that the plane wave exists inside a duct system, the sound wave propagation along the X-direction would be governed by the classical acoustic wave equation as:(1)$$\frac{\partial^{2}p}{\partial x^{2}}=\frac{1}{c_{0}^{2}}\frac{\partial^{2}p}{\partial t^{2}}$$
where *p* is the acoustic pressure, *c*_0_ = 343 m/s represents the sound speed in the air, and *t* is the time. Assuming that the wave is harmonic in time, the sound pressure and particle velocity could be solved as:(2)$$p(x,t)=p_{I}e^{i(\omega t-kx)}+p_{R}e^{i(\omega t+kx)}$$
(3)$$u(x,t)=\frac{p_{I}}{\rho_{0}c_{0}}e^{i(\omega t-kx)}-\frac{p_{R}}{\rho_{0}c_{0}}e^{i(\omega t+kx)}$$
where $i=\sqrt{-1}$ is the imaginary unit, $\rho_{0}$ = 1.204 kg/m^3^ represents the air density, $p_{I}$ and $p_{R}$ are complex pressure amplitudes indicating acoustic waves that propagate along with two opposite directions, ω is the angular frequency, and $k=\omega /c_{0}$ is the wave number.

The transfer matrix method (TMM) has been widely used to evaluate the noise attenuation performance of the mufflers. The transfer matrix of a circular duct of uniform cross-sectional area and length, e.g., from point C to point D in Figure 1d, is given by:(4)$$\left[\begin{array}{c} p(0,t)\\ \rho_{0}c_{0}u(0,t) \end{array}\right]=\left[\begin{array}{l} \begin{array}{cc} T_{11} & T_{12} \end{array}\\ \begin{array}{cc} T_{21} & T_{22} \end{array} \end{array}\right]\left[\begin{array}{c} p(L_{EC},t)\\ \rho_{0}c_{0}u(L_{EC},t) \end{array}\right]={Τ}_{CD}\left[\begin{array}{c} p(L_{EC},t)\\ \rho_{0}c_{0}u(L_{EC},t) \end{array}\right]$$
where *L_EC_* is the length from point C to point D. Sound pressure p(0,t) and $p(L_{EC},t)$ as well as volume velocity u(0,t) and $u(L_{EC},t)$ can be solved via Equations (2) and (3):(5)$$\begin{array}{ll} p(0,t) & =[p_{I}+p_{R}]e^{i\omega t}\\ p(L_{EC},t) & =[p_{I}e^{-ikL_{EC}}+p_{R}e^{ikL_{EC}}]e^{i\omega t}\\ \; & =[(p_{I}+p_{R})\cos kL_{EC}-i(p_{I}-p_{R})\sin kL_{EC}]e^{i\omega t} \end{array}$$
(6)$$\begin{array}{ll} u(0,t) & =\frac{1}{\rho_{0}c_{0}}[p_{I}-p_{R}]e^{i\omega t}\\ u(L_{EC},t) & =\frac{1}{\rho_{0}c_{0}}[p_{I}e^{-ikL_{EC}}-p_{R}e^{ikL_{EC}}]e^{i\omega t}\\ \; & =\frac{1}{\rho_{0}c_{0}}[(p_{I}-p_{R})\cos kL_{EC}-i(p_{I}+p_{R})\sin kL_{EC}]e^{i\omega t} \end{array}$$

By ignoring the time-harmonic terms, Equations (5) and (6) could be re-arranged to a matrix form:(7)$$\left[\begin{array}{c} p(L_{EC})\\ \rho_{0}c_{0}u(L_{EC}) \end{array}\right]=\left[\begin{array}{l} \begin{array}{cc} \cos kL_{EC} & -i\sin kL_{EC} \end{array}\\ \begin{array}{cc} -i\sin kL_{EC} & \cos kL_{EC} \end{array} \end{array}\right]\left[\begin{array}{c} p(0)\\ \rho_{0}c_{0}u(0) \end{array}\right]$$

Equation (7) could be used to determine the sound pressure and particle velocity transmitted through length *L* inside a uniform cross-section duct. Then, the transfer matrix **T_CD_** of Equation (4) could be obtained by inverting Equation (7):(8)$$\left[\begin{array}{c} p(0)\\ \rho_{0}c_{0}u(0) \end{array}\right]=\left[\begin{array}{l} \begin{array}{cc} \cos kL_{EC} & i\sin kL_{EC} \end{array}\\ \begin{array}{cc} i\sin kL_{EC} & \cos kL_{EC} \end{array} \end{array}\right]\left[\begin{array}{c} p(L_{EC})\\ \rho_{0}c_{0}u(L_{EC}) \end{array}\right]=T_{CD}\left[\begin{array}{c} p(L_{EC})\\ \rho_{0}c_{0}u(L_{EC}) \end{array}\right]$$

TMM could also solve the transfer matrix at conjunction points. According to the statement of pressure equality and conservation of volumetric flow, the transfer matrix from point B to point C in Figure 1b is given by:(9)$$\left[\begin{array}{c} p(B)\\ \rho_{0}c_{0}u(B) \end{array}\right]=\left[\begin{array}{cc} 1 & 0\\ 0 & \frac{S_{EC}}{S_{N}} \end{array}\right]\left[\begin{array}{c} p(C)\\ \rho_{0}c_{0}u(C) \end{array}\right]=T_{BC}\left[\begin{array}{c} p(C)\\ \rho_{0}c_{0}u(C) \end{array}\right]$$
where *S_EC_* is the area of duct CD, and *S_N_* is the area of neck AB.

By calculating the product of the transfer matrix in each subsystem, the sound pressure and particle velocity between points A and F in Figure 1b are given by:(10)$$\begin{array}{c} \left[\begin{array}{c} p(A)\\ \rho_{0}c_{0}u(A) \end{array}\right]=T_{T}\left[\begin{array}{c} p(F)\\ \rho_{0}c_{0}u(F) \end{array}\right]=\left[\begin{array}{l} \begin{array}{cc} T_{T11} & T_{T12} \end{array}\\ \begin{array}{cc} T_{T21} & T_{T22} \end{array} \end{array}\right]\left[\begin{array}{c} p(F)\\ \rho_{0}c_{0}u(F) \end{array}\right]\\ \;=T_{AB}T_{BC}T_{CD}T_{DE}T_{EF}\left[\begin{array}{c} p(F)\\ \rho_{0}c_{0}u(F) \end{array}\right] \end{array}$$
where **T_T_** represents the transfer matrix for the side-branch tube of the ITEC, and **T_AB_** to **T_EF_** represents the transfer matrix of each cascaded subsystem. The transfer matrix from **T_AB_** to **T_EF_** could be easily obtained by referring to Equations (8) and (9).

### 2.2. Transfer Matrix of the ITEC

The sound transmission characteristics inside the side-branch tube are pre-requisites to acquiring the sound pressure and particle velocity of the whole duct system. This indicates that the transfer matrix between points *L* and *R* of the main duct is required. The continuous conditions of pressure equilibrium and conservation of volume velocity at the junction position yield:(11)$$\left\{\begin{array}{l} p(L)=p(R)=p(A)=p(F)\\ S_{M}u(L)+S_{N}u(F)=S_{M}u(R)+S_{N}u(A) \end{array}\right.$$

Re-arranging Equation (11), we can obtain:(12)$$u(L)=\frac{S_{N}}{S_{M}}(u(A)-u(F))+u(R)$$

The relationship between U(A) and U(F) in Equation (12) could be derived from Equations (10) and (11):(13)$$\left\{\begin{array}{l} p(A)=p(F)\\ p(A)=T_{T11}p(F)+T_{T12}\rho_{0}c_{0}u(F)\\ \rho_{0}c_{0}u(A)=T_{T21}p(F)+T_{T22}\rho_{0}c_{0}u(F) \end{array}\right.$$

Equation (13) along with Equations (12) and (11) could be solved to determine the transfer matrix between points *L* and *R*:(14)$$\begin{array}{l} \left[\begin{array}{c} p(L)\\ \rho_{0}c_{0}u(L) \end{array}\right]=T_{M}\left[\begin{array}{c} p(R)\\ \rho_{0}c_{0}u(R) \end{array}\right]=\left[\begin{array}{cc} T_{M11} & T_{M12}\\ T_{M21} & T_{M22} \end{array}\right]\left[\begin{array}{c} p(R)\\ \rho_{0}c_{0}u(R) \end{array}\right]\\ =\left[\begin{array}{cc} 1 & 0\\ \frac{S_{N}}{S_{M}}\frac{T_{T12}T_{T21}+(T_{T22}-1)(1-T_{T11})}{T_{T12}} & 1 \end{array}\right]\left[\begin{array}{c} p(R)\\ \rho_{0}c_{0}u(R) \end{array}\right] \end{array}$$

Finally, the transmission loss of the whole duct system could be expressed as:(15)$$TL=20\log_{10}|\frac{p(L)}{p(R)}|=20\log_{10}|\frac{1}{2}(T_{M11}+T_{M12}+T_{M21}+T_{M22})|$$

## 3. Results and Discussion

### 3.1. Validation of Theoretical Prediction

The finite element method (FEM) simulations are performed by commercial software COMSOL Multiphysics to validate the accuracy of the transmission loss results of the analytical model. The parameters of the ITEC model in this validation case are listed in Table 1. The transmission loss results of the frequency domain between 1 and 1000 Hz are shown in Figure 2. It can be seen that the ITEC has three resonance frequencies in the selected frequency domain. The TMM results match the FEM results well, especially near the first transmission loss peak. The frequencies corresponding to transmission loss peaks are listed in Table 2. It could be seen that the transmission loss peaks have a maximum error of 10 Hz near the second peak and a minimum error of 4 Hz near the first peak. In general, the analytical model has relatively high accuracy.

The analytical model could also be used to predict the transmission loss of the infinity tube. According to Lato et al. [19], for an infinity tube, as shown in Figure 1c, the transmission loss is:(16)$$TL=20\log_{10}|\frac{2(S_{M}-S_{2}+S_{M}e^{ikL_{2}}+S_{2}e^{ikL_{2}})}{1+2S_{M}e^{ikl_{2}}}|$$
where *S_M_* represents the cross-section area of the main duct, *S*_2_ denotes the cross-section area of IT, and *L*_2_ represents the length of IT. In Figure 1b, if *S_EC_* has the same value as *S_N_*, the ITEC would become an infinity tube. Therefore, the analytical model of the ITEC could also be used to predict the transmission loss of IT. In this case, matrices **T_BC_** and **T_DE_** in Equation (10) turn into identity matrices, which indicates that Equation (10) becomes:(17)$$T_{T}=\left[\begin{array}{cc} T_{T11} & T_{T12}\\ T_{T21} & T_{T22} \end{array}\right]=T_{AB}T_{CD}T_{EF}$$

Using the expression of **T_T_** in Equation (17), the transmission loss of the infinity tube could be deduced. In the following research, the analytical model based on Equation (15) is used to calculate the transmission loss of IT and then is compared with the results from Equation (16).

As shown in Figure 3, the solid lines represent the transmission loss of ITs with *L*_2_ = 1.15 m and various *S*_2_*/S_M_* ratios. At the same time, ITs with the same geometries are used to examine the transmission loss by Equation (15). Figure 3 illustrates a good agreement between the analytical model from the second part and from the research conducted by Lato et al. [19]. The results indicate that Equation (15) could predict the IT transmission loss.

### 3.2. Noise Attenuation Ability of the ITEC

A comparison of the transmission loss between the IT and ITEC is carried out to examine the noise attenuation ability of the ITEC. The parameters of the ITEC are the same as the geometric model of Table 1, and the IT parameters are selected as the *S*_2_*/S_M_* = 1/4 case in Figure 3, which indicates that IT has the same cross-section area as the neck of the ITEC.

Figure 4 shows the analytical transmission loss between the IT and ITEC. The transmission loss peaks of the ITEC are 115, 403, and 733 Hz, while the transmission loss peaks of IT are 149, 447, and 745 Hz. The results show that an expansion chamber could lead to a decrease of 34, 46, and 12 Hz in resonance frequency. On the other hand, compared with IT, the noise attenuation bands of the ITEC under three transmission loss peaks are non-uniform. In the lower frequency (1–350 Hz), the attenuation band of the ITEC is significantly wider than IT. In medium frequency (350–650 Hz), they are approximately close to each other. In the higher frequency (650–1000 Hz), the attenuation band of the ITEC is narrower than IT. This feature indicates that the ITEC is more efficient in reducing low-frequency noise. In addition, since the ITEC has decreased resonance frequency, it has an advantage in low-frequency noise control compared with IT.

The Helmholtz resonator is widely used as a muffler for ductwork systems in industry [8]. To further examine the noise attenuation ability and assess the potential in the industrial application of the ITEC, a comparison of the transmission loss between the ITEC and Helmholtz resonator system is conducted here. As illustrated in Figure 5a, if a sharable sidewall is placed at the midpoint of the ITEC, the whole system could be regarded as two curved Helmholtz resonators mounted on the same cross-section of the main duct. Cai and Mak [21] have examined the transmission loss of parallel HRs system, which is shaped as shown in Figure 5b. Figure 5a,b indicate that curved HRs are geometrically similar to the parallel HRs system. However, the ductwork system is always located in a limited space. A curved HRs system could save more space if it has the same cavity volume as a straight HRs system.

In this study, the parameters of curved HRs are the same as the ITEC in Table 1. The neck parameters of parallel HRs are *L_N_* = 95.91 mm and *S_N_ =* 1418.6 mm^2^. The cavity volume of parallel HRs is the same as half of the chamber volume of curved HRs, which is easy to obtain from Table 1. As shown in Figure 6, the transmission loss of curved HRs is the same as the ITEC, with the same resonance frequencies and noise attenuation bandwidths. This indicates that the sharable sidewall has no impact on the noise attenuation mechanism of the ITEC. However, the transmission loss of the parallel HRs system is different. It has only two resonance frequencies from 1 to 800 Hz, while the ITEC and curved HRs have three. In addition, under the lower-frequency domain (1–350 Hz), the resonance frequency of parallel HRs is 133 Hz, which has an increase of 14 Hz compared with the ITEC and curved HRs; under the moderate frequency domain (350–650 Hz), the resonance frequency of parallel HRs is 473 Hz, which has an increase of 64 Hz compared with the ITEC and curved HRs. The characteristic of resonance frequencies shows that the ITEC and curved HRs are entirely different from the parallel HRs system, although they are geometrically similar. The ITEC has a lower resonance frequency than parallel HRs, which indicates that the ITEC is more suitable for reducing low-frequency noise. Furthermore, the curved shape of the ITEC and curved HRs system has an advantage in a constrained space. These characteristics indicate that the ITEC would have potential in ductwork systems.

### 3.3. Parametric Study of the ITEC

In this section, ITECs with different geometric parameters are analyzed to discuss the influence of geometrics on noise attenuation performance. Figure 7 shows the transmission loss results of ITECs with different length ratios. The total length of ITECs (*L_EC_* + 2*L_N_*) is fixed (1150 mm), while the neck and expansion chamber lengths have different values. The length values are shown in Table 3. Both FEM simulation and TMM analysis are conducted to validate the accuracy of transmission loss results. According to research in the previous parts, the ITEC would have three peaks of 1–350 Hz, 350–650 Hz, and 650–1000 Hz, respectively. Therefore, the frequency domains are divided into three sub-domains to distinguish 1st, 2nd, and 3rd TL peaks. Under the lower-frequency domain, the peaks of ITECs with different length ratios are close, while they have more significant differences under moderate and higher-frequency domains. We could summarize the following principles for ITECs with different length ratios:

(1)Under the lower-frequency domain, length ratios have little influence on resonance frequency and attenuation bandwidth. ITECs with a higher length ratio would slightly decrease transmission loss peaks and have slightly narrower attenuation bands.(2)Under the moderate frequency domain, the length ratio significantly influences transmission loss performance. ITECs with higher length ratios have a higher resonance frequency and narrower attenuation bands. Compared with the low-frequency condition, ITECs have significantly narrower attenuation bands under the moderate frequency domain, which indicates that ITECs with higher length ratios are not suitable for medium-frequency noise attenuation.(3)Length ratio has a significant influence on transmission loss under the higher-frequency domain. With the length ratio changing from 1/10 to 1/4, the transmission loss peak has increased by nearly 100 Hz. In addition, ITECs with a length ratio equal to 1/4 have a significantly wider bandwidth than the other length ratios, which indicates that the increase in neck length of the ITECs would be good for high-frequency noise attenuation.

Furthermore, the influence of the cross-section area ratio is investigated. As listed in Table 4, three *S_N_/S_EC_* ratios correspond to three different expansion chamber radii and a fixed neck radius. The transmission loss results are shown in Figure 8. It could be obtained from Figure 8 that a higher cross-section area ratio would be better for low-frequency noise attenuation. The ITEC with *S_N_/S_EC_* = 1/4 has the lowest peak frequency and widest noise attenuation bandwidth under the lower-frequency domain. On the contrary, the lower cross-section area ratio would be better for high-frequency noise attenuation. The ITEC with *S_N_/S_EC_* = 1/1.44 has the highest peak frequency and widest noise attenuation band under the higher-frequency domain.

In Figure 9, we perform the transmission loss results of the ITECs with different total lengths. The total length of the ITECs is changed from 0.6 and 0.75 times to the original length (*L* = *L_EC_* + 2*L_n_
*= 1.15 m); *L_EC_* and *L_n_* are also scaled down simultaneously, whereas the radii of the neck and expansion chamber have remained unchanged. As shown in Figure 9, the shorter total length would have a broader sound attenuation bandwidth. At the same time, the transmission loss peak would shift to the higher-frequency domain, even exceeding 1000 Hz, the upper limit of this research. In addition, the TMM results of 0.6 *L* would lead to a more significant error than the FEM results, which is due to the fact that the neck length of 0.6 *L* is very short. According to Ingard [22], an end correction is non-negligible for the aperture neck to improve transmission loss accuracy. For this reason, TMM in this study is not suitable for the short *L_n_* case. Both TMM and FEM results show that the ITECs with shorter lengths would have a broader noise attenuation band. Meanwhile, the transmission loss peaks tend to shift to the higher-frequency domain. Therefore, the ITECs with shorter lengths would have better noise attenuation performance but are not suitable for low-frequency noise reduction. The transmission loss peak (*TLmax*) and the resonance frequency (*f_0_*) of ITECs with different geometric parameters are summarized in Table 5.

Figure 10 illustrates the influence of different geometric parameters of ITEC on the *TLmax*. It can be seen that changing the length ratio leads to a change of 19 dB in the *TLmax* in higher-frequency domain and a change of 11.8 dB in the *TLmax* in moderate frequency domain. Changing the cross-section area ratio has a change of 12.1 dB in higher-frequency domain. Changing the total length has a change of 7.8 dB in lower-frequency domain. Therefore, adjusting the length ratio and the cross-section area ratio are beneficial for improving the higher- and medium-frequency noise attenuation ability, and adjusting the total length is useful for improving the lower-frequency noise attenuation ability. Figure 11 illustrates the influence of different geometric parameters of ITEC on *f_0_.* Changing the total length has more of a significant impact on the resonance frequency than changing the length ratio and the cross-section area ratio. The *f_0_* under three peaks has an increase of 98, 302, and 254 Hz. This indicates that adjusting the total length is an effective way to control the frequency of noise reduction.

## 4. Conclusions

This paper conducts a thorough theoretical and numerical investigation of an innovative noise attenuation device, the ITEC. The conclusions are summarized as follows:

A closed-form equation for the transmission loss of the ITEC device has been derived. The analytical model is compared with the FEM model to validate the accuracy of the transmission loss results. In addition, the analytical model is used to predict the transmission loss of the IT device. Transmission loss results indicate that IT could be regarded as a particular case of the ITEC.

The transmission loss results of the ITEC are compared with those of IT, which shows that the ITEC is more suitable for reducing low-frequency noise than IT devices. The transmission loss results of the ITEC are compared with those of curved HRs and parallel HRs systems. The results show that the ITEC has the same transmission loss results as those of curved HRs system, which indicates that the sharable sidewall does not affect the noise attenuation characteristics of the ITEC device. The ITEC has 14 and 64 Hz resonance frequency reduction than parallel HRs, which indicates that the ITEC is more suitable for reducing low-frequency noise. In addition, the geometry of the ITEC shows that it has an advantage in a constrained space, which indicates that the ITEC would have potential in ductwork systems.

A parametric study is conducted to investigate the influence of geometric parameters on the noise attenuation performance of the ITEC. Transmission loss results of the ITECs with different length ratios, cross-section area ratios, and total length are conducted by TMM and FEM. The transmission loss results could provide guidance on choosing the geometric parameters of the ITECs to reduce duct noise.

## Figures and Tables

**Figure 1 sensors-23-00305-f001:**
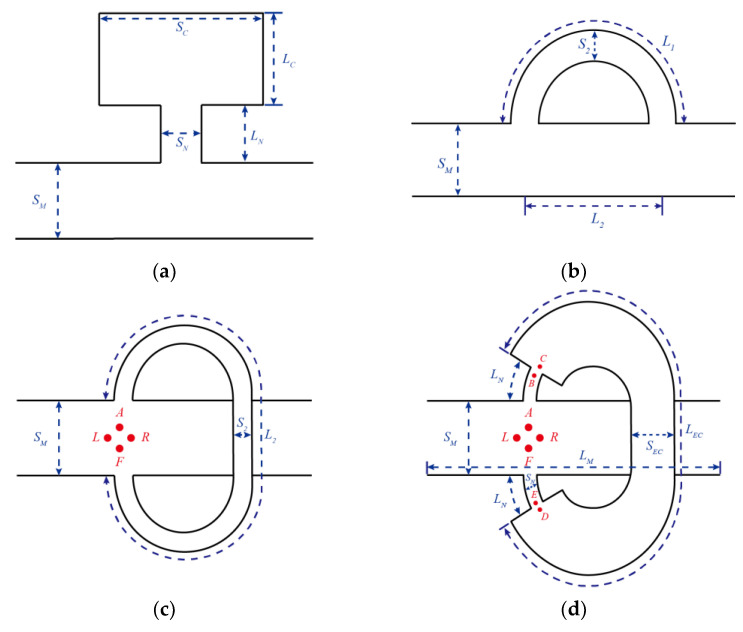
Schematic of the Helmholtz resonator, HQ tube, infinity tube, and infinity tube with an expansion chamber. (**a**) Helmholtz resonator (HR); (**b**) HQ tube; (**c**) infinity tube (IT); (**d**) infinity tube with an expansion chamber (ITEC).

**Figure 2 sensors-23-00305-f002:**
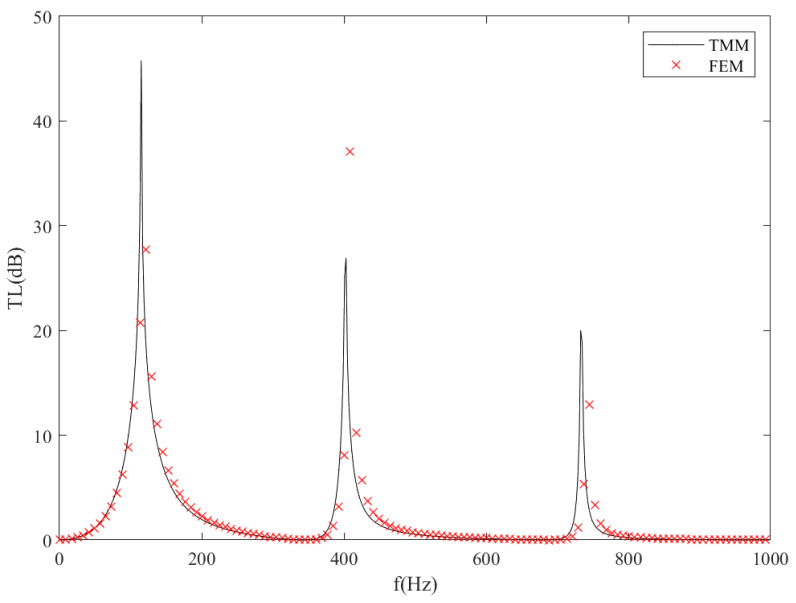
Comparison between the theoretical and numerical results of transmission loss spectra of the ITEC tube; the solid lines represent theoretical predictions by TMM, and the dotted crosses represent simulation results.

**Figure 3 sensors-23-00305-f003:**
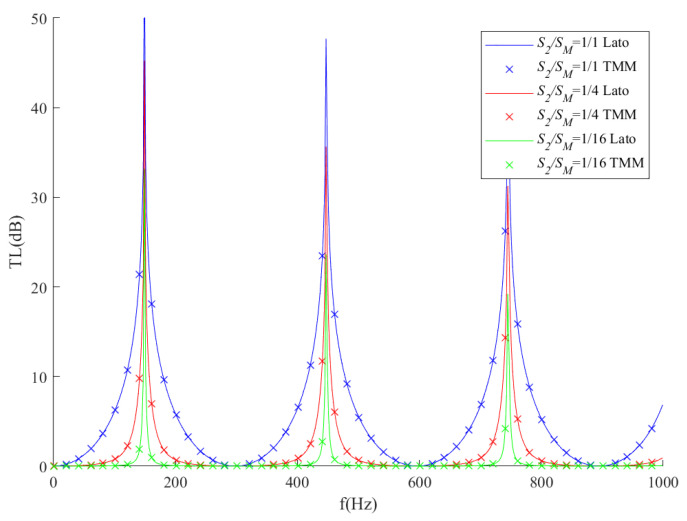
Comparison between the transmission loss of IT based on the analytical model from Section 2 and Equation (16); the solid lines are transmission loss results extracted from Lato et al. [19], and the dotted crosses are calculated by TMM from Section 2.

**Figure 4 sensors-23-00305-f004:**
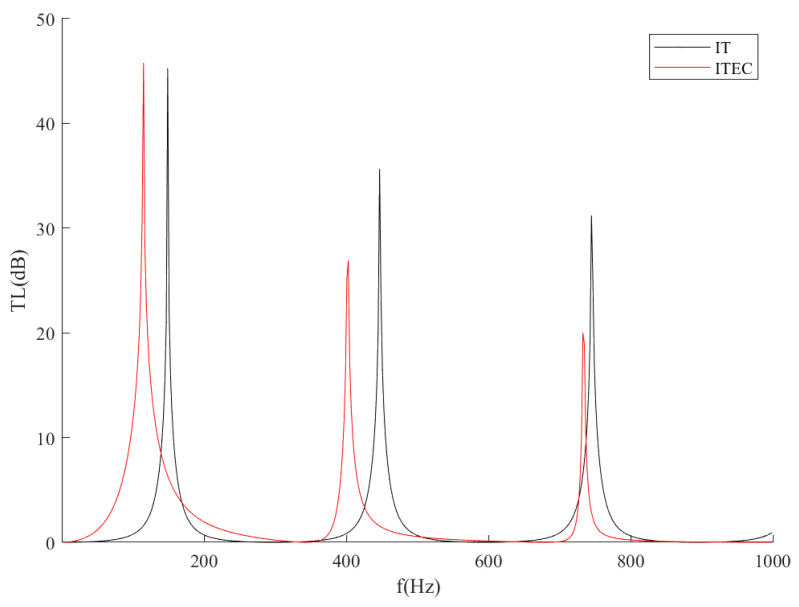
Comparison of transmission loss between the IT and ITEC; the black lines represent IT results, and red lines represent ITEC results.

**Figure 5 sensors-23-00305-f005:**
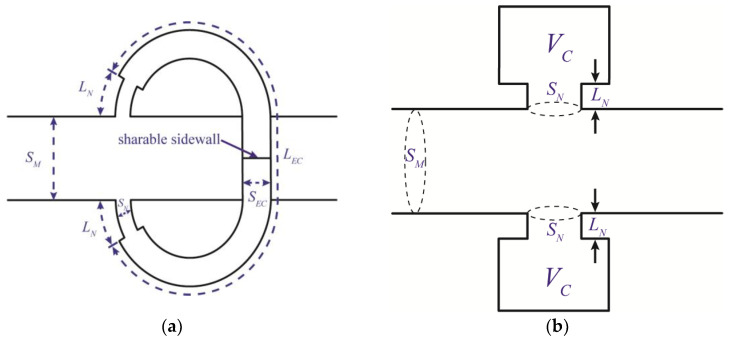
Configuration of curved HRs and parallel HRs systems. (**a**) Curved HRs; (**b**) parallel HRs.

**Figure 6 sensors-23-00305-f006:**
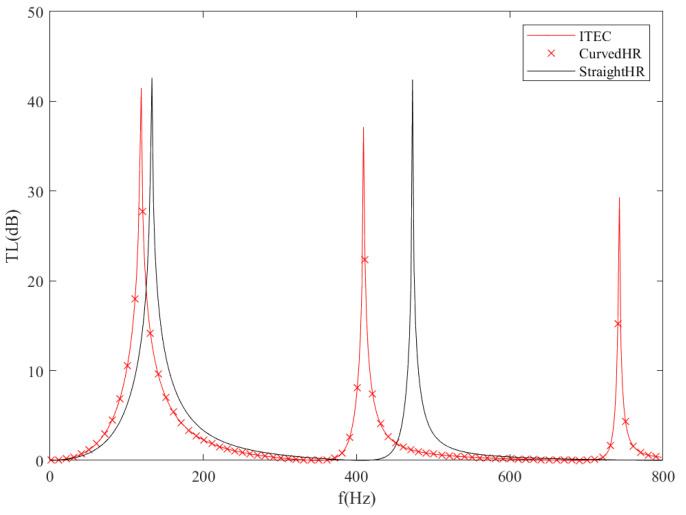
Comparison of the simulated transmission loss results between the ITEC and the curved HRs and parallel HRs systems; the red lines represent the ITEC results, black lines represent parallel HRs system results, and the red dotted crosses represent curved HRs system results.

**Figure 7 sensors-23-00305-f007:**
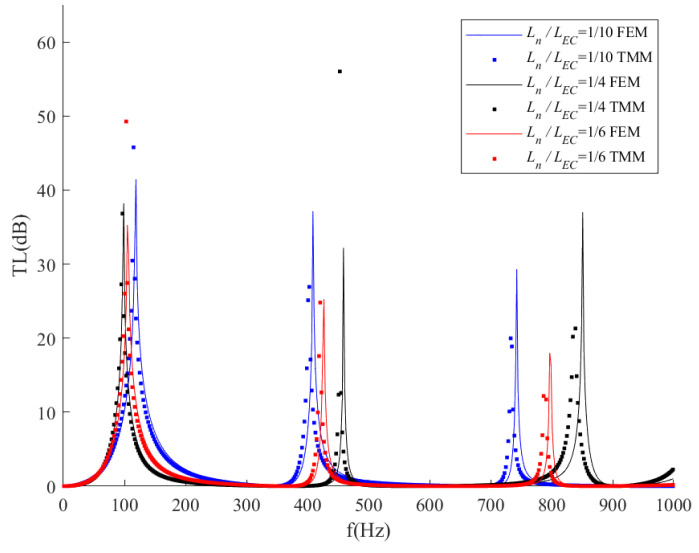
Comparison of transmission loss with respect to different length ratios of the expansion chamber.

**Figure 8 sensors-23-00305-f008:**
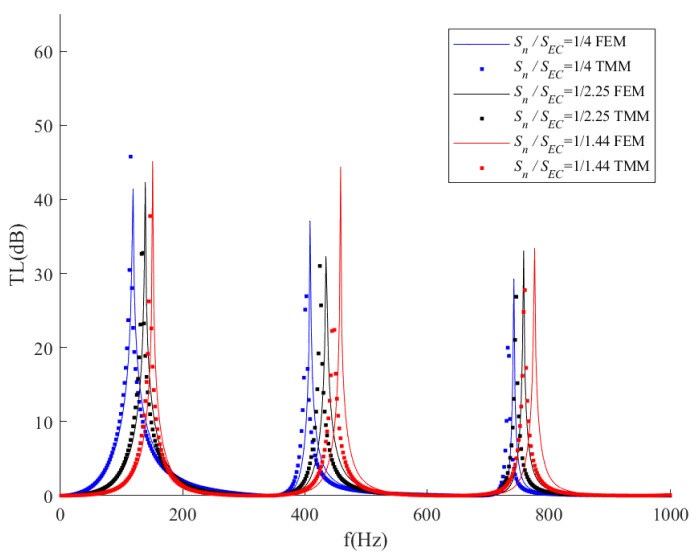
Comparison of transmission loss with respect to different cross-section area ratios of the expansion chamber.

**Figure 9 sensors-23-00305-f009:**
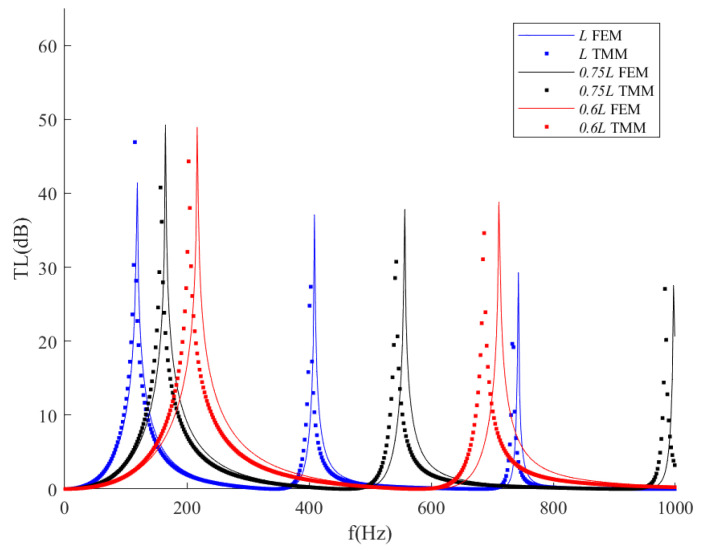
Comparison of transmission loss with respect to different lengths of the ITEC.

**Figure 10 sensors-23-00305-f010:**
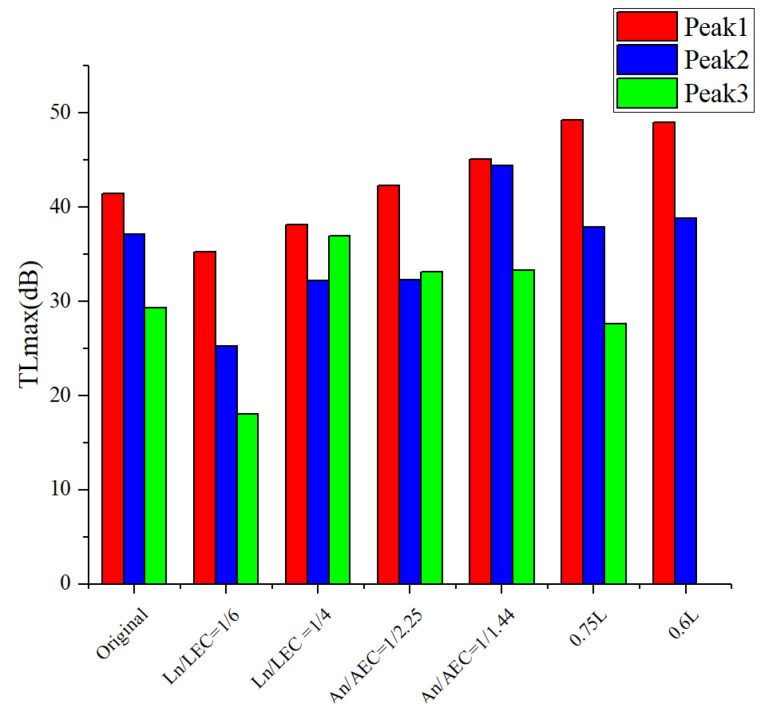
Variation of the transmission loss peak of the ITEC with different geometric parameters.

**Figure 11 sensors-23-00305-f011:**
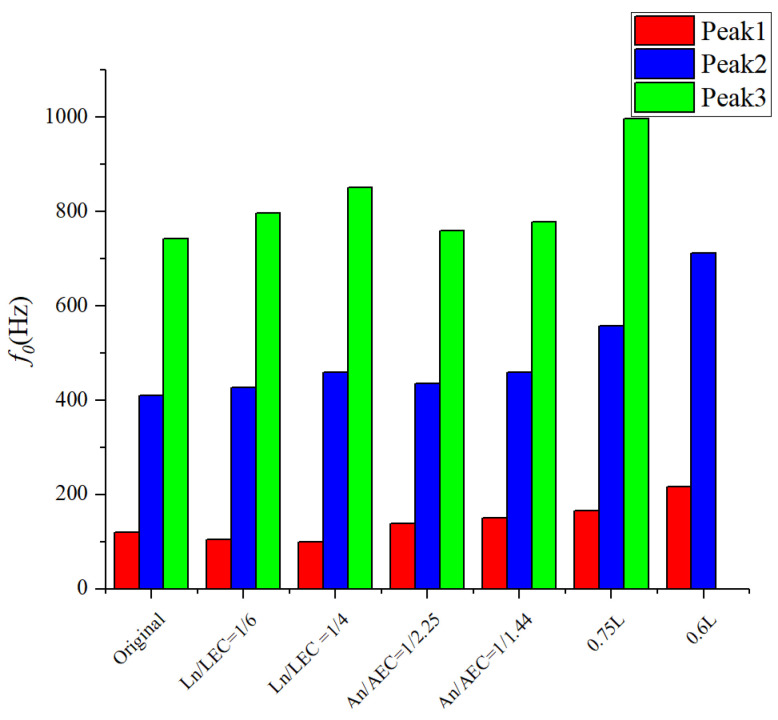
Variation of the resonance frequency of the ITEC with different geometric parameters.

**Table 1 sensors-23-00305-t001:** Parameters of the ITEC model for simulation.

	Length (mm)	Area (mm^2^)
Main duct	*L_M_ *= 1000	*S_M_ *= 5674.5
Neck	*L_N_ *= 95.91	*S_N_ *= 1418.6
Expansion Chamber	*L_EC_ =* 958.19	*S_EC_ *= 5674.5

**Table 2 sensors-23-00305-t002:** The frequencies correspond to the transmission loss peak.

	TMM	FEM
1st peak	115 Hz	119 Hz
2nd peak	403 Hz	409 Hz
3rd peak	733 Hz	743 Hz

**Table 3 sensors-23-00305-t003:** Parameters of *L_N_* and *L_EC_* of ITECs in Figure 7.

*L_N_/L_EC_*	*L_N_*	*L_EC_*
1/10	95.91 mm	958.19 mm
1/6	143.75 mm	862.5 mm
1/4	191.65 mm	766.67 mm

**Table 4 sensors-23-00305-t004:** Parameters of *S_N_* and *S_EC_* of the ITECs in Figure 8.

*S_N_/S_EC_*	*R_N_*	*R_EC_*
1/1.44	21.25 mm	25.5 mm
1/2.25	21.25 mm	31.875 mm
1/4	21.25 mm	42.5 mm

**Table 5 sensors-23-00305-t005:** Summary of the transmission loss peak and the resonance frequency of ITEC with different geometric parameters.

	Peak 1		Peak 2		Peak 3	
	*TLmax* (dB)	*f_0_ *(Hz)	*TLmax* (dB)	*f_0_ *(Hz)	*TLmax* (dB)	*f_0_ *(Hz)
*L_n_/L_EC_* = 1/10	41.40	119.00	37.06	409.00	29.27	743.00
*L_n_/L_EC_* = 1/6	35.20	105.00	25.20	427.00	18.01	797.00
*L_n_/L_EC_* = 1/4	38.10	99.00	32.20	459.00	36.90	851.00
*S_n_/S_EC_ *= 1/4	41.43	119.00	37.06	409.00	29.27	743.00
*S_n_/S_EC_* = 1/2.25	42.30	139.00	32.30	435.00	33.10	759.00
*S_n_/S_EC_* = 1/1.44	45.10	151.00	44.40	459.00	33.30	777.00
*L*	41.40	119.00	37.06	409.00	29.27	743.00
0.75 *L*	49.22	165.00	37.86	557.00	27.60	997.00
0.60 *L*	48.97	217.00	38.79	711.00	-	-

## Data Availability

Not applicable.

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
