# Peer review of "An Infinity Tube with an Expansion Chamber for Noise Control in the Ductwork System"

_sensors, 2022, doi:10.3390/s23010305_

Round 1

Reviewer 1 Report

The detailed comments are listed in the remarks

  •  

Author Response

Reply to Reviewer 1*:

*Reviewer comments are indicated in black, our responses in blue, and our text additions in red. All modifications in the manuscript have been highlighted.

  1. It is better to add nomenclature after the abstract or the conclusion part.

Response: Thanks for your advice. We have added a nomenclature chart after the conclusion part.

  1. For duct noise, when the frequency is higher than a certain value, according to the Tyler & Sofrin mode theory, there will be multiple non-plane waves propagating in the duct. What impact will appear on the research in this paper? Whether the acoustic propagation model with only plane waves assumed in formula (1) continues to be applicable?

Response: We gratefully appreciate your valuable comments. In this research, we used a circular duct as the main duct. The cut-off frequency of the main duct can be calculated by:

where fc0 is the cut-off frequency, D is the diameter of the main duct, c0 is the sound speed. The diameter of the main duct is 42.5cm. After calculation, the cut-off frequency is around 4729Hz. Acoustic waves with higher modes cannot propagate when the frequency of the sound source is under this frequency range. Therefore, only plane waves exist in the whole system. There won’t be multiple non-plane waves in our study.

  1. The comparison results in figure 2 show that the frequency is in good agreement, but the maximum difference of TL value at the peak is 25dB. What impact will this have on the subsequent research?

Response: Thanks for pointing this out. The mismatch of the TL peaks appeared between the TMM and FEM. The frequency range in our study is between 1Hz and 1000Hz. We used TMM to calculate TL at the interval of 1Hz. However, it will consume a lot of computational resources if we use the same frequency interval to calculate TL by FEM simulation. Therefore, we used 5Hz as the frequency interval when conducting FEM simulation. Because of the large frequency interval, the TL peak can't be obtained. Consequently, we compared peak frequency to validate the accuracy of our analytical method. We confirm the comparison results don't impact our analytical and numerical methods.

  1. The image is not clear enough and needs to be replaced with a higher resolution resulting image.

Response: Thanks for pointing this out. The resolution of our figures was reduced when the manuscript was converted to PDF. We have replaced the low-resolution figures in manuscript and uploaded all of the high-resolution figures in a single zip archive.

  1. Is this low, medium and high frequency range a common definition in the industry or sensor investigation?

Response: The low, medium, and high-frequency range is a common industry definition. In this study, we used this classification method to distinguish the 1st, 2nd, and 3rd resonance peaks.

  1. It needs to be explained how the dB value of the whole paper is calculated, whether it is sound power level or sound pressure level. In the acoustic noise reduction design, 19dB is a huge energy difference. According to the difference of 25dB at the peak shown in Figure 2, will the change of 19dB be greatly discounted?

Response: In this study, we investigated the transmission loss (TL) performance of the ITEC. TL is defined as , where pin and pout are the acoustic pressure level of the incident and reflected acoustic wave. Therefore, the dB value of the whole paper was represented by the sound pressure level. We have made a revision in equation(10) to clarify this. In question 2, we have explained the reason that caused the 25dB TL difference between TMM and FEM in fig.2. In Table 5, all of the TL values were obtained from TMM. We confirm that the 19dB difference isn't discounted.

[Line113]: Finally, the transmission loss of the whole duct system could be expressed as:

  1. It is suggested that the size of figure abscissa and ordinate in the whole paper should be unified. The ordinate of Figure 10 is obviously larger than that of Figure 9. There are similar problems in other pictures in this paper.

Response: As suggested by the reviewer, we have revised all the figures to ensure they have a unified coordinate.

  1. The reference format should be uniform. The format of the references should be according to the journal's requirements. Most of the references are more than 20 years old. In order to increase the timeliness of the paper, could you please add relevant literatures in recent years so as to more accurately track the research progress?

Response: Thanks for pointing this out. We have modified the reference style to ensure it fits the journal’s requirements. Besides, we have presented relevant literature in recent years in the Introduction part:

[Line45-50]: “…Kim et al. [15] designed a virtual HQ tube system to achieve the desired transmission loss performance under a required frequency range. Wang et al. [16] combined HQ tube with micro-perforated panels and developed a new noise control device. Ahmadian et al. [17] developed a genetic algorithm to optimize the parameters of a HQ tube. Mazzaro et al. [18] numerically investigated air flow movement inside the HQ tubes…”

  1. the manuscript can be accepted for publication after minor revision.

Response: Thank you for the positive comment.

Reviewer 2 Report

This paper presents the acoustic analysis of an infinity tube with an expansion chamber and shows the effects of geometric parameters on the transmission loss characteristics. The transmission losses were calculated by transfer matrix and the calculation was compared with a FEM simulation. Although the results are not at all surprising, the paper is clear and concise, and the reviewer recommends publication in the journal Sensors after consideration of the following points.

  1. The authors validated the results of TMM with those of FEM simulations. However, usually, TMM has higher accuracy than the FEM in the low-frequency range because it directly solves the wave equation, except for waveguides with short concatenate connections or small constrictions. In contrast, FEM has its own computational errors, and its accuracy also should be validated. Therefore, it sounds strange to me for a comment saying that the analytical model has relatively high accuracy at Line 123. If there were two-dimensional distributions in FEM results and it cannot be modeled by 1D TMM, it should be mentioned that.
  2. In the introduction, adding a figure of comparison for the Helmholtz resonator, HQ tube, infinity tube, and ITEC geometries might be helpful for the readers.
  3. Some dots (sentence period) for the end of equations are missing (e.g. Eq. (6) and Eq. (12))
  4. The reviewer cannot read Eq. (13) in the PDF.
  5. Point “B” in Fig. 1 (a) should be “F” to match with Fig. 1 (b). It was confusing because there is another point B in Fig. 1 (b) at a different position.
  6. Where is L_M in Fig. 1?
  7. What are the values of L_EC and S_EC for Fig. 2? Those should be listed in Table 1.
  8. Peak1 to Peak3 should be 1st peak, 2nd peak, and 3rd peak in Table 2.
  9. A1, A2, and l2 should be replaced with S_M, S_2, and L2 to match those with the figures in this paper.
  10. Line 152: equation (2.15) should be (15)?
  11. Line 210: total length is 2L_N + L_EC? Mentioning it here will be helpful.
  12. The values in Table 5 should have units (dB and Hz?). In addition, what is Fre? If you want to use a variable for frequency, you can just write italic f.

Author Response

Reply to Reviewer 2*:

*Reviewer comments are indicated in black, our responses in blue, and our text additions in red. All modifications in the manuscript have been highlighted.

This paper presents the acoustic analysis of an infinity tube with an expansion chamber and shows the effects of geometric parameters on the transmission loss characteristics. The transmission losses were calculated by transfer matrix and the calculation was compared with a FEM simulation. Although the results are not at all surprising, the paper is clear and concise, and the reviewer recommends publication in the journal Sensors after consideration of the following points.

Response: Thank you for your feedback to help improve our paper. We appreciate the time and input of your review. Please find our responses to your comments below.

  1. The authors validated the results of TMM with those of FEM simulations. However, usually, TMM has higher accuracy than the FEM in the low-frequency range because it directly solves the wave equation, except for waveguides with short concatenate connections or small constrictions. In contrast, FEM has its own computational errors, and its accuracy also should be validated. Therefore, it sounds strange to me for a comment saying that the analytical model has relatively high accuracy at Line 123. If there were two-dimensional distributions in FEM results and it cannot be modeled by 1D TMM, it should be mentioned that.

Response: Thank you for this insight. As the reviewer stated, the 1D TMM can solve the wave equation directly. However, the 1D model can't predict the 2D acoustic pressure distributions. For example, the acoustic pressure near the conjunction of the main duct and the ITEC can't be solved from the 1D model, since the TMM ignored the boundary effect on the acoustic wave. In fact, the discontinuity between the ITEC and the main duct can influence the acoustic wave propagation inside the narrow neck of the ITEC. Therefore, the simulation results of 2D FEM are more accurate. FEM can introduce numerical errors. Many researchers have used FEM to validate their analytical results of acoustic wave propagation inside a waveguide, such as https://doi.org/10.1121/1.5097167, https://doi.org/10.1016/j.ast.2018.03.002, and https://doi.org/10.1121/1.5139886. We think our simulation results can be used to validate the analytical model.

  1. In the introduction, adding a figure of comparison for the Helmholtz resonator, HQ tube, infinity tube, and ITEC geometries might be helpful for the readers.

Response: Thank you for pointing this out. We have added the schematic of HR and HQ tube on Fig.1.

(a)

(b)

(c)

(d)

Figure 1. Schematic of the Helmholtz resonator, HQ tube, infinity tube and infinity tube with an expansion chamber (a) Helmholtz resonator (HR), (b) HQ tube; (c) infinity tube (IT), (d) infinity tube with an expansion chamber (ITEC).

  1. Some dots (sentence period) for the end of equations are missing (e.g. Eq. (6) and Eq. (12)).

Response: We have carefully checked all the equations in the manuscript. The format of the equations meets the journal's requirements now.

  1. The reviewer cannot read Eq. (13) in the PDF.

Response: Eq.(13) has been added after Line110.

  1. Point "B" in Fig. 1 (a) should be "F" to match with Fig. 1 (b). It was confusing because there is another point B in Fig. 1 (b) at a different position.

Response: As suggested by the reviewer, Point "B" is changed to "F" in Fig.1(a) (Now Fig.1(c)).

  1. Where is L_M in Fig. 1?

Response: Thank you for pointing this out. LM represents the length of the main duct. We have added LM in Fig.1.

  1. What are the values of L_EC and S_EC for Fig. 2? Those should be listed in Table 1?

Response: LEC and SEC are the lengths and the cross-sectional areas of the expansion chamber. Their values have been added in Table 1.

Table 1. Parameters of the ITEC model for simulation.

Length (mm)

Area (mm2)

Main duct

LM=1000

SM=5674.5

Neck

LN=95.91

SN=1418.6

Expansion Chamber

LEC=958.19

SEC=5674.5

  1. Peak1 to Peak3 should be 1st peak, 2nd peak, and 3rd peak in Table 2.

Response: We have changed Table 2 accordingly.

  1. A1, A2, and l2 should be replaced with S_M, S_2, and L2 to match those with the figures in this paper.

Response: Thank you for pointing this out. We have changed these parameters accordingly.

[Line138-140]: According to Lato et al.[19], for an infinity tube, as shown in Figure 1(c), the transmission loss is:

where SM represents the cross-section area of the main duct, S2 denotes the cross-section area of IT, and L2 represents the length of IT."

  1. Line 152: equation (2.15) should be (15)?

Response: We changed the manuscript accordingly.

  1. Line 210: total length is 2L_N + L_EC? Mentioning it here will be helpful.

Response: We agreed with the reviewer and rewrote the sentence.

[Line]: "…The total lengths of ITECs (LEC+2LN) are fixed (1150mm)…"

  1. The values in Table 5 should have units (dB and Hz?). In addition, what is Fre? If you want to use a variable for frequency, you can just write italic f.

Response: Thank you for pointing this out. We have added units in Table 5. Fre represents the resonance frequency. In the manuscript, we changed all of the Fre to f0.

Table.5. Summary of the transmission loss peak and the resonance frequency of ITEC with different geometry parameters.

Peak1

Peak2

Peak3

TLmax (dB)

f0 (Hz)

TLmax (dB)

f0 (Hz)

TLmax (dB)

f0 (Hz)

Ln/LEC =1/10

41.40

119.00

37.06

409.00

29.27

743.00

Ln/LEC =1/6

35.20

105.00

25.20

427.00

18.01

797.00

Ln/LEC =1/4

38.10

99.00

32.20

459.00

36.90

851.00

An/AEC=1/4

41.43

119.00

37.06

409.00

29.27

743.00

An/AEC=1/2.25

42.30

139.00

32.30

435.00

33.10

759.00

An/AEC=1/1.44

45.10

151.00

44.40

459.00

33.30

777.00

L

41.40

119.00

37.06

409.00

29.27

743.00

0.75L

49.22

165.00

37.86

557.00

27.60

997.00

0.60L

48.97

217.00

38.79

711.00

-

-
